# Genome-Wide Identification of bHLH Transcription Factor Family in *Malus sieversii* and Functional Exploration of *MsbHLH155.1* Gene under *Valsa* Canker Infection

**DOI:** 10.3390/plants12030620

**Published:** 2023-01-31

**Authors:** Shanshan Jia, Xiaojie Liu, Xuejing Wen, Abdul Waheed, Yu Ding, Gulnaz Kahar, Xiaoshuang Li, Daoyuan Zhang

**Affiliations:** 1National Key Laboratory of Ecological Security and Sustainable Development in Arid Areas, Urumqi 830000, China; 2College of Resources and Environment, University of Chinese Academy of Sciences, Beijing 100000, China; 3Xinjiang Key Laboratory of Conservation and Utilization of Plant Gene Resources, Xinjiang Institute of Ecology and Geography, Chinese Academy of Sciences, Urumqi 830000, China; 4Turpan Eremophytes Botanical Garden, Chinese Academy of Sciences, Turpan 838000, China

**Keywords:** *Malus sieversii*, bHLH transcription factor, genome identification, expression analysis, disease resistance

## Abstract

Xinjiang wild apple (*Malus sieversii*) is an ancient relic; a plant with abundant genetic diversity and disease resistance. Several transcription factors were studied in response to different biotic and abiotic stresses on the wild apple. Basic/helix–loop–helix (bHLH) is a large plant transcription factor family that plays important roles in plant responses to various biotic and abiotic stresses and has been extensively studied in several plants. However, no study has yet been conducted on the bHLH gene in *M. sieversii*. Based on the genome of *M. sieversii*, 184 putative *MsbHLH* genes were identified, and their physicochemical properties were studied. *MsbHLH* covered 23 subfamilies and lacked two subfamily genes of *Arabidopsis thaliana* based on the widely used classification method. Moreover, *MsbHLH* exon–intron structures matched subfamily classification, as evidenced by the analysis of their protein motifs. The analysis of cis-acting elements revealed that many *MsbHLH* genes share stress- and hormone-related cis-regulatory elements. These MsbHLH transcription factors were found to be involved in plant defense responses based on the protein–protein interactions among the differentially expressed *MsbHLHs*. Furthermore, 94 *MsbHLH* genes were differentially expressed in response to pathogenic bacteria. The qRT-PCR results also showed differential expression of *MsbHLH* genes. To further verify the gene function of bHLH, our study used the transient transformation method to obtain the overexpressed *MsbHLH155.1* transgenic plants and inoculated them. Under *Valsa* canker infection, the lesion phenotype and physiological and biochemical indexes indicated that the antioxidant capacity of plants could increase and reduce the damage caused by membrane peroxidation. This study provides detailed insights into the classification, gene structure, motifs, chromosome distribution, and gene expression of *bHLH* genes in *M. sieversii* and lays a foundation for a better understanding disease resistance in plants, as well as providing candidate genes for the development of *M. sieversii* resistance breeding.

## 1. Introduction

The basic/helix–loop–helix (bHLH) protein is a transcription factor found in a wide range of animals and plants [1]. It has an alternative basic/helix–loop–helix structural domain with two distinct parts: one for the basic amino acid region and one for the helix–loop–helix region [2,3]. The basic region is involved in the DNA binding to the E-box (usually CANNTG) or G-box (CACGTG) motifs in its target genes. The HLH region consists of two α-helices containing hydrophobic residues, which are necessary for dimerization to alter the expression of the target genes involved in various signaling pathways [4]. The classification of bHLH proteins is usually based on sequence homology within conserved bHLH domains [5]. Based on their sequence homology and phylogenetic relationship, bHLH transcription factors are usually divided into 15–26 groups in plants [6,7,8,9]. The *bHLH* gene family has been investigated in plants such as *Arabidopsis thaliana* (162 members) [10], *Dendrobium officinale* (98 members) [11], *Malus domestica* (188 members) [12], *Oryza sativa* (167 members) [13], *Raphanus sativus* (213 members) [14], *Prunus mume* (95 members) [15], *Cucumis sativus* (142 members) [16], *Hibiscus hamabo* (162 members) [17], and *Populus deltoids* (185 members) [18]. Identification of the *bHLH* gene family not only contributes to the preservation of wild apple gene resources, but can also be helpful in the molecular breeding of apples to improve resistance.

The Xinjiang wild apple (*M. sieversii*) is a relic from the third century. The *M. sieversii* is highly likely to be the ancestor of modern cultivated apples (*M. domestica*) [19]. The *M. sieversii* forest is mainly distributed in the TianShan Mountains of Kazakstan and Kyrgyzstan of central Asia and Xinjiang, China, and is considered a wild apple gene pool with important scientific research value [20]. In addition, *M. sieversii* is an ideal material for improving the disease resistance of apples in scion breeding [21]. However, the long disturbances of inappropriate anthropic activities and extreme climate change have caused outbreaks of pests and diseases, and the number and distribution area of *M. sieversii* populations in Xinjiang have been shrinking [20,22]. Hence, it is extremely urgent to rescue and preserve the Tianshan Wild Fruit Forest. At present, *Hsp20* (12 members) [23], *Hsp90* (8 members) [24], and *WRKY* (112 members) [25] gene families have been identified in *M. sieversii.* However, the screening identification of functional gene resources in *M. sieversii* is not sufficient. As one of the largest transcription factors in plants, *bHLH* genes play important roles in plant growth and development [26], material synthesis [27,28], and stress resistance [29,30]. In addition to providing a theoretical basis for the molecular breeding of *M. sieversii* and *M. domestica,* the identification of bHLH gene families in *M. sieversii* will contribute to understanding of important functional genes.

The bHLH transcription factors play important roles in plant disease resistance [31]. For example, overexpression of the *CmbHLH87* gene in tobacco may increase resistance to powdery mildew, bacterial wilt, and scab [32]. The bHLH transcription factor Gmpib1 in soybean (*Glycine max*) plays a crucial role in defense against *Phytophthora* root rot [33]. It was also found that MdbHLH92 plays an active role in the salicylic acid (SA)-mediated powdery mildew defense pathway [34]. Two bHLH transcription factors (bHLH3 and bHLH6) could activate downstream genes in the resistance process of the tomato to bacterial spots caused by *Xanthomonas gardneri*, thereby participating in the resistance pathway [35]. Overexpression of *GhbHLH171* in cotton (*Gossypium hirsutum*) activated jasmonic acid (JA) biosynthesis and signaling pathways and improved the plant resistance to *Verticillium* wilt [36]. Research has shown that transcription factors bind to cis-acting elements in stress-responsive gene promoters and control gene expression by regulating plant defense against various abiotic and biological stresses [37]. Rice MAPK OsMPK3 is a TEY-type TEY-type enzyme that phosphorylates OsbHLH65, a transcription factor bound to the E-box element that responds to biotic stresses and defense-related hormones. The existing studies indicate that bHLH transcription factors may play a role in JA- and SA-mediated disease resistance pathways. [38]. Studies have also shown that exogenous SA can induce the expression of JA biosynthetic genes in the early stage of infection, leading to the accumulation of resistant proteins and other substances to defend against fungi [39].

However, no research has been reported for the bHLH family in *M. siseversii*. We identified the bHLH gene family in *M. sieversii* and investigated its expression pattern in response to *Valsa* canker infection. In addition, *M. sieversii* cultured seedlings overexpressing *MsbHLH155.1* under *V. mali* infection were tested in a gene-resistant functional identification system based on transient transformation. In order to improve *M. sieversii’s* disease resistance, this study provides an essential theoretical basis and high-quality candidate genes for molecular breeding and improving the strain.

## 2. Results

### 2.1. Identification of bHLH Transcription Factors in M. sieversii

In order to identify the *MsbHLH* family genes in *M. sieversii*, 201 putative MsbHLH protein sequences were obtained. By using the Simple Modular Architecture Research Tool (SMART) database and CD-Search (Conserved Domains) database to identify candidate genes with “HLH” domains and removing redundant sequences, we identified 184 sequences as genes in the bHLH family of *M. sieversii*. According to their sequence similarity and phylogenetic relationship with a single AtbHLH protein, they were named MsbHLH001-MsbHLH162.6. Since one *AtbHLH* gene may have multiple homologous *MsbHLH* genes, the group of 184 *bHLH* genes was named “*MsbHLH001*–*MsbHLH162.6*”. Sequence analysis revealed that the lengths of the MsbHLH protein sequences varied from 92 to 879 aa and had predicted molecular weights of 28837.95–90283.67 kDa. The predicted isoelectric points (pIs) ranged from 4.6 to 10.37. The instability indexes of nine MsbHLH proteins were less than 40, suggesting that these putative MsbHLH proteins have good stability. The instability indexes of the remaining bHLH proteins were more than 40, suggesting unstable proteins. The subcellular localization prediction results showed that MsbHLH transcription factors were mainly located in the nucleus, and nuclear localization signals also showed that 138 MsbHLH proteins had nuclear localization signals, of which 54 were monomolecular nuclear localization signals (MN), 29 were bimolecular nuclear localization signals (BN), and 55 were MN and BN types (Appendix A).

### 2.2. Analysis of the Phylogenetic Relationships, Gene Structure, and Conserved Motifs of MsbHLHs

Previous studies have shown that bHLH transcription factor families in plants were generally divided into 15–32 subfamilies [9,40]. To confirm the structural characteristics of the MsbHLH proteins, 184 MsbHLH protein sequences were analyzed by multiple sequence alignment. All 184 MsbHLH protein sequences contain characteristic bHLH regions: two helix regions, one loop region, and one basic region (Figure 1). Furthermore, the conserved amino acids in the bHLH domain with a sequence consistency greater than 50% were highlighted in yellow (Figure 1A). To create sequence markers, 184 amino acid sequences from the MsbHLH homologous domain were used. The MsbHLH proteins in *M. sieversii* contain 17 conserved amino acids of the bHLH domain. The amino acid residues Arg-5, Arg-6, Leu-16, and Leu-43 in 184 putative MsbHLH proteins are highly conserved, as shown in Figure 1B.

To understand the evolutionary relationship of bHLH transcription factors among species, a neighbor-joining phylogenetic tree was constructed using 159 *Arabidopsis* bHLH sequences and 184 *M. sieversii* bHLH sequences (Figure 2). The genes of these two species were divided into 25 subgroups (S1–S25), and the *bHLH* gene of *M. sieversii* was distributed in 23 of 25 branches (subgroups). In order to further understand the phylogenetic relationship, we evaluated the number of representative genes in bHLH subgroups of *A. thaliana* and *M. sieversii*. There were no members of the S1 and S15 in *M. sieversii*, indicating that these genes may be missing during evolution. The largest MsbHLH subfamily was S25, with 20 members, whereas the smallest subfamily, S20, had only 2 members.

Conserved motif prediction showed that 10 motifs were determined in 184 MsbHLH protein sequences of *M. sieversii* (Figure 3A). It was confirmed that almost all sequences of MsbHLH contained conserved motifs 1 and 2, whereas other conserved motifs were only found in certain gene sequences. In addition, the corresponding positions of each bHLH protein motif were also conserved.

It has been observed that putative MsbHLH proteins within a subfamily often display similar motif patterns. Motifs 1, 2, 3, and 8 were evident in every member of S3; motifs 1, 2, 3, and 8 were evident in every member of S4; motifs 1, 2, 3, and 8 were evident in every member of S17; and motifs 1, 2, 4, and 9 were evident in every member of S18. Furthermore, we found that motif 5 was detected in S7, S8, S9, S10, and S12; and motif 6 was detected only in S8, S9, and S10.

Additionally, we investigated the exon–intron structure of 184 MsbHLH protein sequences (Figure 3B) in order to understand their gene structure characteristics. As expected, the putative MsbHLH proteins from the same subfamily seem to have similar exon–intron tissues, and a total of 66 members had no untranslated region (UTR).

### 2.3. Chromosomal Distribution of MsbHLH Genes

Based on annotation information from the genome data of *M. sieversii*, 184 *MsbHLH* genes were found on its 17th chromosome (Figure 4). *M. sieversii* has an uneven distribution of 184 *bHLH* genes on its 17 chromosomes. The most *MsbHLH* genes were found on chromosomes 1 and 11. There were 17 *MsbHLH* genes on chromosome 2. The lowest number of *MsbHLH* genes was in chromosome 8, which only contained five *MsbHLHs*. It has been shown that tandem duplicates and segmental duplicates play crucial roles in the expansion of gene families and the emergence of new gene functions [41]. We found three tandem duplication events in *M. sieversii*, *MsbHLH21.1* and *MsbHL21.2* on chromosome 2; *MsbHLH10.2* and *MsbHLH10.3* on chromosome 9; and *MsbHLH38.2* and *MsbHLH39* on chromosome 16. It is likely that these genes, which formed tandem duplication events, belong to the same subfamily and have similar gene structures.

### 2.4. The Cis-Regulatory Elements and Protein–Protein Interaction Analysis of MsbHLHs

In order to predict how *MsbHLH* genes respond to biotic stress and their regulatory relationships, 2000 bp promoter regions were analyzed for 184 *MsbHLH* genes. These functional elements were divided into five categories: the G-box of the *bHLH* gene, hormone-responsive elements, abiotic-stress-related elements, biotic-stress-related elements, and plant-growth-related elements. There were 167 *MsbHLH* genes with G-box elements, accounting for 90.7% of the total number of genes. Among them, *MsbHLH060.1* and *MsbHLH035.1* transcription factors contain the most elements, with 13 each. In addition to hormone-response elements (ABRE, TGA element, P-box, TGACG motif), *MsbHLH* genes contain abiotic-stress-response elements (ARE, MBS, STRE, TC-rich repeats, LTR). Among them, there are 29 hormone-response elements in the promoter region of *MsbHLH035.1.* The promoter region of *MsbHLH084.3* contains 17 abiotic-stress-response elements. Among the results of the study, family members play a key role in hormone action pathways and stress response to abiotic stress. Moreover, 165 *MsbHLH* promoters contained biological stress-related components (W-box, WUN-motif, WRE3), accounting for 89.7% of the total *bHLH* genes; 143 *MsbHLH* promoter regions were rich in JA-related *cis*-elements; and 82 MsbHLH promoter regions were rich in SA-related elements. In addition, 104 *MsbHLH* promoter regions contained a W-box; 85 *MsbHLH* promoter regions contained WER3 elements; and 102 *MsbHLH* promoter regions contained a WUN-motif, indicating that they may be involved in a series of biological stress responses and defenses (Appendix A). Plant-growth-related elements (Circadian, CAT-box, As-1) were found in 181 *MsbHLH* promoters (Figure 5), indicating that the family of genes may play roles in plant growth.

To identify potential interacting proteins with the MsbHLH transcription factors, a protein–protein interaction (PPI) network generated with STRING was used to predict the interaction relationship of 94 putative MsbHLH proteins (Figure 6A). Many MsbHLH proteins interacted with more than one MsbHLH. MsbHLH008.3 (homologous to plant-photosensitive pigment AtPIF3) and MsbHLH015.2 (homologous to AtPIL5) contributed to the hypocotyl growth of plants and regulated anthocyanin synthesis. In addition, MsbHLH042.1 (AtTT8) was a regulatory factor that regulated the flavonoid pathway in synergy with TT1, PAP1, and TTG1, involving proanthocyanidins and anthocyanin biosynthesis by affecting the expression of dihydroflavonol-4 reductase gene (*DFR*). Interestingly, MYC2 is involved in the jasmonic-acid-mediated pathway in this protein network when interacting with multiple JASMONATE-ZIMDOMAIN (JAZ) proteins. Moreover, MsbHLH155.1 was found to interact with LBA1 (AT5G47010) and multiple cell-wall-related proteins (AT1G05310) by predicting its protein network (Figure 6B). These results suggest that these MsbHLH transcription factors may interact with JAZ proteins and are involved in a jasmonic-acid-mediated signaling pathway.

### 2.5. Expression Patterns of MsbHLH Genes during Valsa Canker Disease

To further determine the involvement of *MsbHLH* genes in response to *V. mali* infection, the expression pattern of differentially expressed *MsbHLH* genes was analyzed from the transcriptome data [42]. Infection with *V. mali* resulted in the quantification of 94 *MsbHLH* genes by mapping reads (Figure 7A), and the expression pattern was very complex, which was roughly divided into four clusters. There was continuous up-regulation of the first cluster’s expression for 0–5 days, continuous down-regulation of the second cluster’s expression for 0–5 days, continuous up-regulation of the third cluster’s expression on days 0–1 and 2–5, and constant down-regulation of the last cluster’s expression for 0–1 days and constant up-regulation of the last cluster’s expression for 2–5 days. As a result of a pathogenic infection, the expression pattern of the *MsbHLH* genes may differ, suggesting that their response times and pathways are different. To verify the reliability of transcriptome data, nine differentially expressed genes (DEGs) (*MsbHLH* genes) were selected for qRT-PCR verification. As a result, all genes except *MsbHLH092.1* exhibited differential expression patterns that were consistent with transcriptome data (Figure 7B), indicating that the qRT-PCR data are consistent with the Illumina data. Ultimately, *M. sieversii* may respond differently to *V. mali* infection by up- or down-regulating *MsbHLH* genes.

### 2.6. The MsbHLH155.1 Confers a Canker Disease-Resistance Function

A transient expression system based on *M. sieversii* seedlings was employed to investigate the disease-resistance functions of *MsbHLH* genes. *MsbHLH155.1* was transiently overexpressed in *M. sieversii*. The expression level of *MsbHLH155.1* was detected by qRT-PCR. Plants transiently overexpressing *MsbHLH155.1* exhibited significantly higher levels of expression than plants transiently overexpressing the empty vector control group (Figure 8A). Compared with the control plants, the average lesion area of *MsbHLH155.1* overexpression plants decreased 0.0076 cm^2^ on the 2nd day and 0.1121 (cm^2^) on the 3rd day (Figure 8B,D). The content of bacteria decreased by 0.2412 (copies/mL) compared with the control group (Figure 8E). In addition, the malondialdehyde (MDA) content decreased by 5.571 (ugꞏg^−1^ꞏFW) (Figure 8C). The H_2_O_2_ content also decreased by 112.31 (ugꞏg^−1^ꞏFW) in the leaves of *MsbHLH155.1* transgenic plants and was significantly decreased after infection with *V. mali* (Figure 8F). As a result, *MsbHLH155.1* increased *M. sieversii*’s antioxidant capacity, reduced damage to cell membranes, inhibited the proliferation and growth of *V. mali*, and preserved cell structure, so it may play a key role in the response to *Valsa* canker disease.

## 3. Discussion

### 3.1. Characterization of the MsbHLH Transcription Factor Family

In plants, the bHLH transcription factor family is the second largest family of eukaryotic transcription factors after MYBs. The genome sequences of *A. thaliana* and *O. sativa* were used in previous phylogenetic analyses of bHLH proteins in plants [13,43,44,45], which provides a useful but limited framework for angiosperm bHLH protein classification.

An increasing number of species have been identified that contain bHLH proteins, and bHLH protein families have been found in all terrestrial plants sequenced to date. In this study, 184 *MsbHLH* genes were identified and characterized. A larger number of MsbHLHs were found in *Arabidopsis* (162 members) [10], but fewer than those in *M. domestica* (188 members) [12]. Due to the expansion of genes in cultivated apples during evolution, *M. sieaersii* might have been the ancestor of *M. domestica.* The 23 subgroups of MsbHLHs were identified based on phylogenetic analysis (Figure 2). Within the bHLH gene family of *M. domestica,* 21 subgroups have been identified [12]. This could be the result of *M. sieversii* having a greater number of subfamilies than *M. domestica*. There are greater diversity and complexity of amino acid sequences in the bHLH transcription factors in *M. sieversii*. Multiple sequence analysis revealed that all 184 MsbHLH proteins had conserved bHLH domains (Figure 1). For example, Leu-16 and Leu-43 amino acids are relatively conserved in the helical dimerization domain. Conservative sequence analysis also revealed that most of the 184 MsbHLH proteins contained conserved motifs 1 and 2, which are also found in *M. domestica* [12]. A phylogenetic relationship between 184 *MsbHLH* genes was further confirmed by gene structure and motif analysis (Figure 3). Thus, these results prove that *M. sieversii’s bHLH* gene is reliable, since all 184 MsbHLHs exhibit bHLH characteristics.

### 3.2. Phylogenetic Analysis and Evolution of MsbHLH Genes

In order to explore the evolutionary relationship between 184 MsbHLH proteins in *M. sieversii* and 159 AtbHLH proteins in *A. thaliana*, a phylogenetic tree was constructed. Twenty-three subfamilies were found in *M. sieversii*, but not subfamilies S2 and S15, indicating that there is a difference in evolutionary processes between *M. sieversii* and *Arabidopsis*. It was observed that the numbers of MsbHLH transcription factors were amplified in many subfamilies, such as S8, S10, and S23. The expansion of numbers of the *bHLH* gene family could improve the environmental adaptability of *M. sieversii*. The MYC2 transcription factor, as a direct inhibitor of the jasmonate ZIM domain (JAZ), plays a role in regulating the JA signaling pathway in multiple biological processes [46,47]. In this study, MsbHLH004.1, MsbHLH004.2, and MsbHLH004.3 were homologous with MYC4. In addition, EGL3 and GL3 played important roles in anthocyanin synthesis in *A. thaliana* [48,49]. AtGL3, AtEGL3, and AtMYC1 can also regulate hairy body formation and root hair pattern [50,51]. In this study, MsbHLH012.1 and MsbHLH012.2 were homologous with AtMYC1. Therefore, MsbHLH012.1 and MsbHLH012.2 may play the same roles in the development of plant roots and the process of anthocyanin synthesis. In our study, MsbHLH008.1, MsbHLH008.2, and MsbHLH008.3 were homologous with AtPIF3, as they are plant pigment interaction factors [52,53]. A number of studies indicate that PIFs not only play a direct role in regulating gene expression but also optimize plant development and growth by interacting with various factors in addition to endogenous signaling pathways (hormones, light, temperature, and circadian rhythm), abiotic signaling pathways (defense response), and biotic signaling pathways (light, temperature, circadian rhythms) [54]. Adaptation occurs in species during the evolutionary process according to their living environment. Since *M. sieversii* lives in a relatively cold environment and suffers from *Valsa spp.* infection, biotic- and abiotic-stress-related genes may be amplified more in the evolutionary process, improving its adaptability.

### 3.3. MsbHLH Genes May Play Important Roles in Plant Disease Resistance

Increasing plant resistances to powdery mildew [32], bacterial wilt [32], scab [32], root rot [33], tomato bacterial spot [35], *Verticillium* wilt [36], and white mold [55] are dependent on bHLH transcription factors. Research on the disease-resistance mechanisms of bHLH transcription factors has revealed that bHLH transcription factors might work via JA or SA signaling. [31]. For example, JA signaling pathway activation by GhbHLH171 overexpression in cotton contributes to *Verticillium’s* wilt resistance. When stress signals are present, JA accumulates, resulting in the degradation of JA-ZIM (JAZ), which then activates the transcription factors inhibited by JAZ, allowing for defense response and development to occur [36].

Some pathogen-related (PR) genes controlled by the SA signaling pathway were markedly down-regulated in rice RNA sequences from OsbHLH61-RNAi plants [31], suggesting a close association between bHLHs and SA signaling pathways. Through SA- and JA-mediated signaling pathways, bHLH transcription factors may regulate the expression of some pathogen-related genes to enhance plant resistance to infection. In *M domestica*, the transcription factor MdERF100 physically interacts with the MdbHLH92 transcription factor to mediate powdery mildew resistance and enhance plant resistance by regulating the JA and SA signaling pathways [34]. However, the function of resistance to pathogens in *MsbHLH* gene family has not been studied. The bHLH genes may also regulate SA or JA in *M. sieversii*, which requires experimental confirmation with more precise tests to explain why *M sieversii* is resistant to *Valsa* canker.

Plant defense responses are initiated by pathogens or pathogen-derived inducers, which lead to extensive transcriptional changes [56,57]. During plant immune responses, a number of defense-related genes are upregulated, and they may play essential roles in disease resistance [58,59]. Based on the previous research, it appears that pathogen-related genes have *cis*-elements in their promoters. Consequently, determining functional *cis*-acting elements in promoters is an essential step to understanding gene function experiments [60]. Pathogenic fungi induce various defense-related genes that are largely described by *cis*-acting elements in the genome, but little is known about their *cis*-activity, including on the W-box [61], S-box [62], and GCC-box [63]. Some of these elements were used to construct synthetic promoters, such as W-box and GCC-box, which contribute to pathogen induction [64]. As shown by *cis*-element analysis, MsbHLH-promoter regions contain elements related to JA, SA, W-box, and WER3, suggesting that they may be involved in the biotic stress response and defense. Additionally, interaction networks were predicted for 94 MsbHLH transcription factors. Several factors have been implicated in the promotion of pathogen-induced callose deposition, including MsbHLH028 (homologous with AtMYC2), SA biosynthesis, PR1 expression, and SA response. In addition, MYC2 is required for the JA-mediated defense against the necrotrophic pathogen *Botrytis cinerea* [65]. As a component of JA-mediated defense, it interacts with proteins such as JAZ1, JAZ9, and CRY2. The study found that the JAZ protein interacts with bHLH (TT8, GL3, and EGL3) and the R2R3 MYB transcription factor (MYB75). This inhibits the JA-mediated accumulation of anthocyanins and the formation of trichomes through the WD-repeat/bHLH/MYB transcription complex [66]. It is inferred that MsTT8, MsGL3, and MsEGL3 might be involved in the JA-regulated anthocyanin accumulation and trichome initiation in *M. sieversii* in response to *V. mali* infection. Furthermore, MsbHLH028 (homologous with MYC2) might be a key transcription factor of the JA signaling pathway in plants. During JA signaling rest, a group of JAZ proteins physically recruit transcriptional co-repressor (TOPLESS) to form a repression complex, inhibiting the expression of JA-responsive genes in MYC2 and its close homologs [67]. MYC2 may also contribute to plant defense by interfacing with the plant stomatal regulator PITY7 (JAZ9). Plant JA-mediated defense may be affected by interactions between CRY2 and MYC2 [65]. According to these results, MsbHLH transcription factors may also participate in the JA signaling and response pathways. Subsequently, we analyzed the *MsbHLH* genes that may respond to biotic stress, which is of great significance for mining disease resistance gene resources in *M. sieversii*.

To investigate the role of *MsbHLH* genes in plant-disease resistance, we analyzed transcriptome sequencing results under pathogen infection conditions. It was found that nearly half of the *MsbHLH* genes responded to pathogen infection. Based on the results, it was found that these differentially expressed *MsbHLH* genes had relatively complex expression patterns in response to *V. mali*. We found that MsbHLH155.1 (homologous to MdbHLH155) participates in the JA pathway to regulate anthocyanin synthesis [68]. Based on the above results, we hypothesized that MsbHLH155.1 may also be involved in JA-mediated signaling pathways to enhance plant defense against pathogens. Reactive oxygen species (ROS) play an integral role as signaling molecules in the regulation of numerous biological processes such as growth, development, and responses to biotic and/or abiotic stimuli in plants [69]. With the measurements of physiological indicators (MDA and H_2_O_2_ levels), the MDA and H_2_O_2_ contents were significantly reduced with the overexpression of *MsbHLH155.1* compared with the control. It was demonstrated that *MsbHLH155.1* presented disease-resistance functions through scavenging reactive oxygen species. In *MsbHLH155*.*1*-transient-overexpression plants, the average areas of lesions in the leaves significantly decreased by suppressing the pathogen’s proliferation. Based on the above results, *MsbHLH155.1* can enhance plant disease resistance. It improves the ability to scavenge ROS, reduces plant cell membrane damage, and inhibits pathogen growth. In summary, these results provide a good reference for further research into the functions of the *MsbHLH* gene family.

## 4. Materials and Methods

### 4.1. Genome-Wide Identification and Classification of bHLH Transcription Factors in M. sieversii

We downloaded the bHLH transcription factor protein sequences of *Arabidopsis* from the plant transcription factor database v4.0 (http://planttfdb.cbi.pku.edu.cn/ (accessed on 22 January 2023)). The hidden Markov model (HMM) file corresponding to the bHLH domain (PF00010) was downloaded from the Pfam protein family database (http://pfam.sanger.ac.uk/ (accessed on 22 January 2023)). Then, HMMER 3.0 software was used to build a hidden Markov model, and the existence of HLH core sequences was verified using the PFAM and SMART programs (http://smart.embl-heidelberg.de/ (accessed on 22 January 2023)). The 184 *MsbHLH* genes were identified in the *M. sieversii* genome. The *M. sieversii* genome (version JAHTLV010000000) was retrieved from the NCBI (https://www.ncbi.nlm.nih.gov/ (accessed on 22 January 2023)) databases [70]. BLASTp was used to verify bHLH proteins. Finally, the basic features of the trihelix proteins of the *bHLH* genes of *M. sieversii* (sequence length, MW, pI, subcellular localization) were identified using the ExPasy (http://web.expasy.org/protparam/ (accessed on 22 January 2023)).

### 4.2. bHLH Gene Structure and Conserved Motif Analysis

To analyze the characteristic domain of MsbHLH proteins, ClustalW was used to generate multi-sequence alignment with default parameters. Mega software (version 6.0) and ESPript 3.0 (https://espript.ibcp.fr/ESPript/cgi-bin/ESPript.cgi (accessed on 22 January 2023)) were then used to adjust the bHLH structure domain while using the deduced amino acid sequences. The online program MEME (http://meme.nbcr.net/meme/intro.Html (accessed on 22 January 2023)) was used to analyze the motifs of the MsbHLH proteins. Then, the motifs, CDS, and UTR areas of *MsbHLH* genes were analyzed using TBtools [71].

### 4.3. Chromosomal Locations, Intron–Exon Structures, and Cis-Element Analysis of MsbHLH Genes

A total of 184 *MsbHLH* genes were mapped from chromosomes of *M. sieversii* based on the location information in the *M. sieversii* genome database in the NCBI. Analysis of the maps was carried out using TBtools [71]. The 2000 bp genomic DNA sequence upstream of the bHLH gene promoter was downloaded and submitted to the predicted cis-elements by the Plant CARE system (http://bioinformatics.psb.ugent.be/webtools/plantcare/html/ (accessed on 22 January 2023)).

### 4.4. Phylogenetic Analysis of the MsbHLH Gene Family

The 159 AtbHLH proteins of *Arabidopsis* were downloaded from TAIR (https://www.arabidopsis.org/ (accessed on 22 January 2023)). The full-length amino acid sequences of *Arabidopsis* and *M. sieversii* were aligned with Clustur W software, and then MEGA 6.0 software was used to obtain a tree using the neighbor-joining method with a bootstrap of 1000 replicates. Finally, the system’s evolutionary tree was beautified using iTOL (https://itol.embl.de (accessed on 22 January 2023)).

### 4.5. Protein Interaction Network Analysis of MsbHLH Transcription Factors

The 94 differentially expressed MsbHLH protein sequences were submitted to the STRING (version 11.5, https://cn.string-db.org (accessed on 22 January 2023)) website. The bHLH protein sequences in *A. thaliana* were chosen as the references, and then the minimum required interaction score (0.4000) was set to middle confidence. Following blast analysis, the orthologous genes with the highest scores were used to construct the network. Genes with no interactions with other genes were removed. Using the blast results, functional annotations were manually pasted.

### 4.6. Expression Patterns of MsbHLH Genes under Valsa mali Infection

Twigs of *M. sieversii* were collected in May 2017 from the area (43°23′2.20′′ N; 83°35′43.48′′ E) in a natural Wild Reserve Forest in Yili, Xinjiang. Following a 75% medical alcohol sterilization, the twigs were injured with a sterilized pattern wheel (2 cm in diameter) and inoculated with a mycelial plug (5 mm) from a 3-day-old canker pathogen, *Valsa mali* (isolate EGI1). After 0, 1, 2, and 5 dpi, barks of twigs near the canker were separately harvested and frozen for RNA extraction. Each sample contained three biological replicates. Bark samples from 0 dpi timepoints were collected for RNA extraction as controls. The differentially expressed *MsbHLH* genes were identified in the transcriptome of *M. sieversii* infected by *V. mali* [42]. The genes were considered to be differentially expressed according to a|log_2_ (fold change)|≥1 with the Q-value < 0.05. The expression pattern heat map was generated using FPKM values of genes by TBtools software [71].

Under *V. mali* infection, the differentially expressed *MsbHLH* genes were selected for qRT-PCR. By blasting primer sequences against the NCBI database, primer specificity for qRT-PCR was evaluated using Primer5 software.(Appendix A). The mRNA was reverse-transcribed into cDNA using the PrimeScript RT reagent Kit with the gDNA Eraser (TAKARA, RR047A, Japan) kit. The qRT-PCR was manipulated by C1000 TM*CF96TM (Bio-Rad, Laboratories, CA, Singapore). The elongation factor α (*EF-α*) gene was used as an internal reference [72]. The 2^−ΔΔCt^ method [73] was used to calculate the relative expression level and conduct three biological repetitions. Statistical analysis was performed using SPSS 26.0 (SPSS Inc., Chicago, IL, USA) software.

### 4.7. Determination of Anti-Valsa Canker Index of MsbHLH155.1 Gene

The intact coding sequences (CDS) of the studied transcription factors were fused in frame with the C-terminus of 3× flag tags under the control of the CaMV 35S promoter in the p1307-flag plant expression vector. The intact CDS of the studied functional genes replaced 5× Myc tags and cloned into the 1307-myc vector under the control of the CaMV 35S promoter to generate the plant expression vectors. All constructs were confirmed by DNA sequencing and transferred into *Agrobacterium tumefaciens* EHA105. The infection process of transient transformation according to the method of Liu et al. [25]. The *M. sieversii* leaves were transiently expressed with *MsbHLH155.1* and p1307-flag (empty vector). The leaves inoculated with *V. mali* isolate EGI1 were photographed at 1, 2, and 3 dpi. The lesion areas were calculated by ImageJ software. The area ratios of the diseased lesion were analyzed by Excel. Determination of the concentrations of H_2_O_2_, MDA, and Bacteroidetes in the samples was performed. The plants transformed with empty p1307-flag or p1307-myc were used as the controls. Statistical analysis was performed using SPSS 26.0 (SPSS Inc., Chicago, IL, USA) software. Three independent biological replicates were performed to ensure the accuracy of the analyses. Each biological replicate contained three technical replicates. The value showed the expression level as the mean ± standard error (*n* = 9). The data were submitted for analysis using Duncan’s test, following one-way ANOVA at the *p* ≤ 0.05 and *p* ≤ 0.01 levels.

## 5. Conclusions

Using the whole genome of *M. sieversii*, the bHLH transcription factor family was identified and characterized. According to the phylogenetic analysis, 184 members of the MsbHLH family were divided into 23 subfamilies, with conservative gene structures and motifs. It was predicted that the MsbHLH domain has certain functional characteristics. Additionally, the transcriptomic data of the *MsbHLH* gene revealed various expression profiles after pathogen infection, which may contribute to a better understanding of the resistance function of the gene. *MsbHLH155.1* was identified as a disease-resistant gene, through inhibiting pathogen growth and scavenging ROS to reduce damage to plant cell membranes. In conclusion, this study provides new insight into the resistance function and regulatory mechanism of the MsbHLH protein in *M. sieversii.*

## Figures and Tables

**Figure 1 plants-12-00620-f001:**
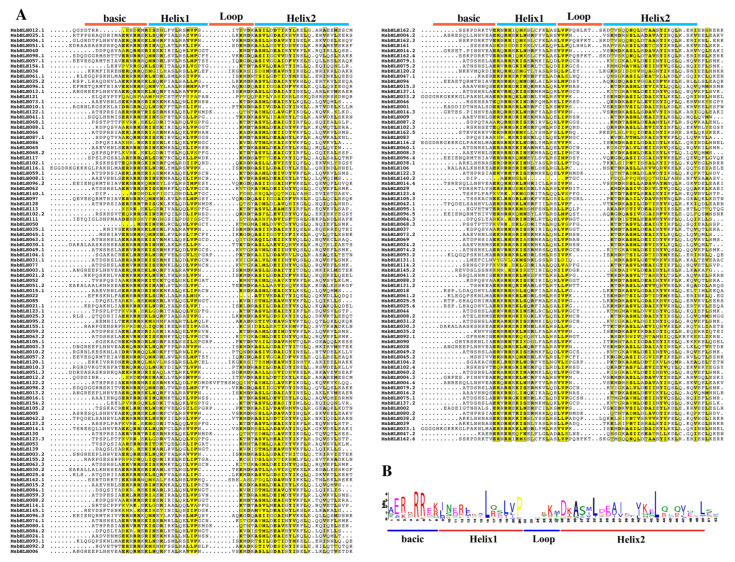
Conserved domains in putative MsbHLH proteins. (**A**) Amino acid sequence alignment analysis of MsbHLH protein. ClustalW and ESPript 3.0 were used to create a multi-sequence alignment project with default parameters of the MsbHLH proteins. The similarity coloring scheme shows % equivalence; global score was 0.7. (**B**) Conservative motif of MsbHLH transcription factor. The MEME online program was used to analyze the motifs of the MsbHLH proteins, the number of motifs was 10, the optimum motif width was from 8 to 60, and other settings were defaults.

**Figure 2 plants-12-00620-f002:**
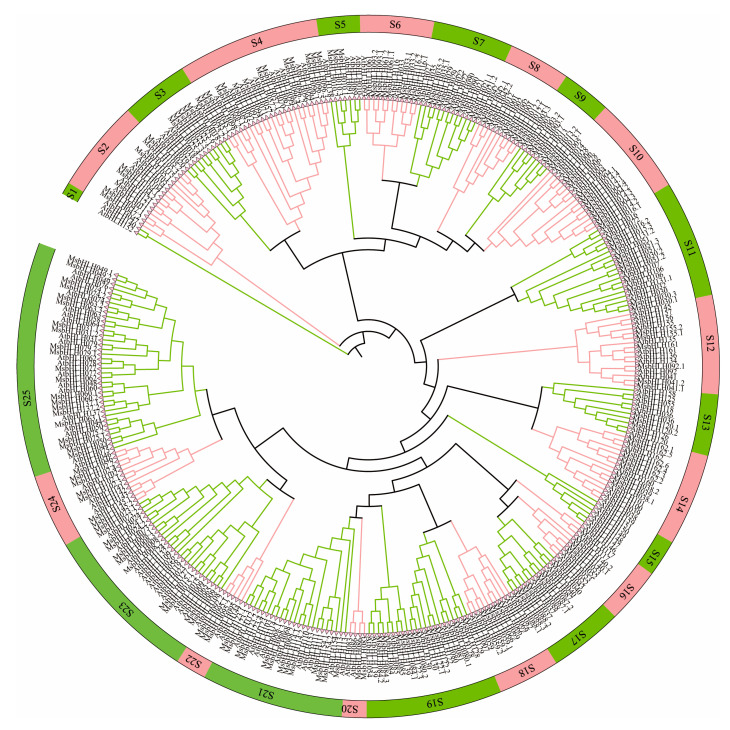
Phylogenetic tree of *M. sieversii* and *A. thaliana.* Phylogenetic tree constructed from the neighbor-joining method using the bHLH transcription factor domain for *M. sieversii.* MEGA 6.0 software to obtain a tree using the neighbor-joining method with a bootstrap of 1000 replicates. S1–S25 represent subfamily 1–subfamily 25.

**Figure 3 plants-12-00620-f003:**
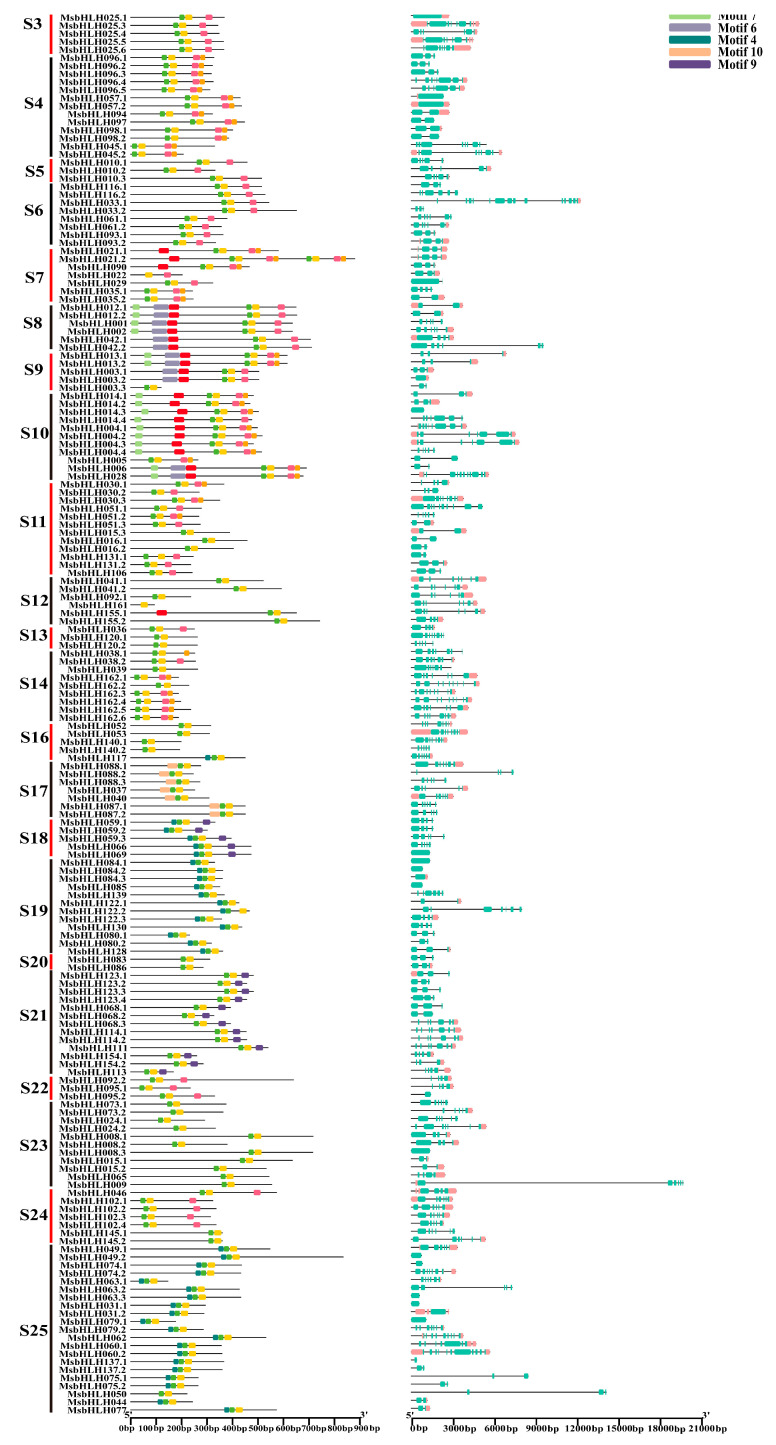
Conserved motif types and distributions in putative MsbHLH proteins. A: Conserved motif analysis of the *M. sieversii* bHLH family. The conserved motifs were identified using MEME software with the following parameters: maximum number of motifs with 10, optimum motif width of 8 to 50, and other settings as defaults. B: Gene structure of the *M. sieversii* bHLH family. The intron–exon structure was obtained based on the gene annotation file of the *M. sieversii* genome and visualized using the TBtools. S2–S25 represent subfamily 2–subfamily 25.

**Figure 4 plants-12-00620-f004:**
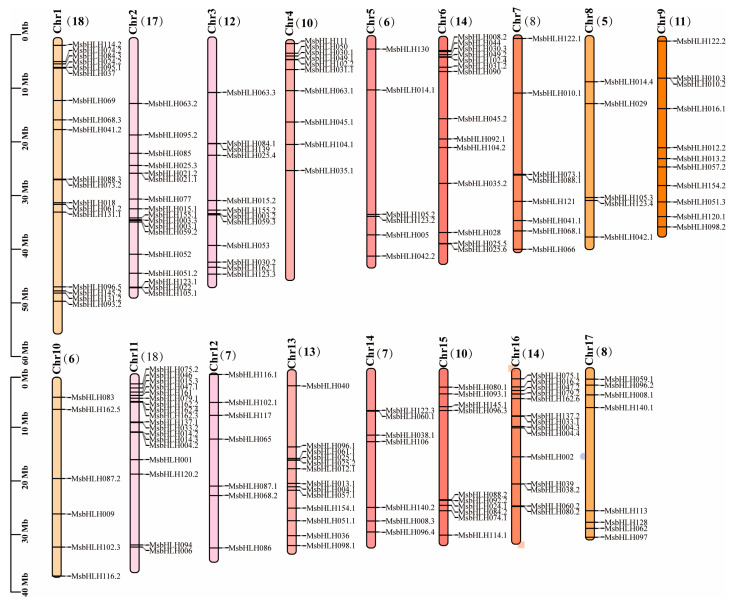
Chromosomal distributions of *MsbHLH* genes. TBtools was used to analyze maps. The name on the side of each chromosome corresponds to the approximate location of each *MsbHLH* genes.

**Figure 5 plants-12-00620-f005:**
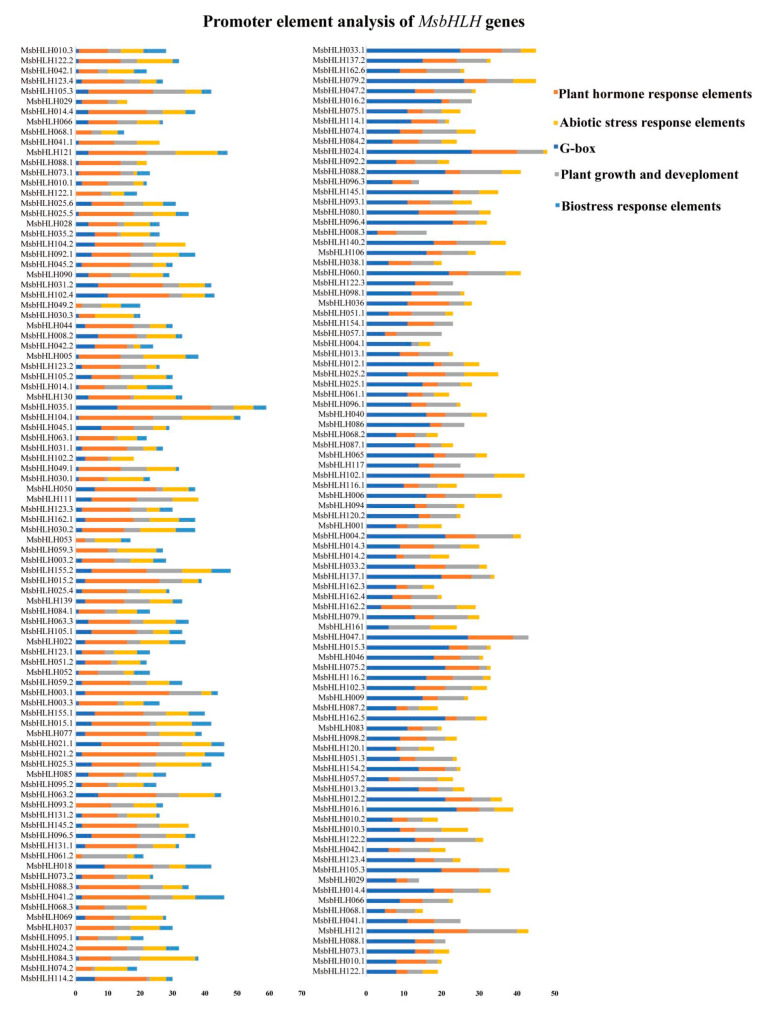
*Cis*-element analysis of 184 *MsbHLH* gene promoters. PlantCARE predicted the potential cis-regulatory elements in the 2000 bp promoter region upstream of *M. sieversii*. Different colors imply elements related to G-box, growth (circadian rhythm control), plant hormones (abscisic acid, auxin, methyl jasmonate, gibberellin, and salicylic acid), biotic stress response (insect pest and plant disease), and abiotic stress (anaerobic induction, low temperature, and drought induction).

**Figure 6 plants-12-00620-f006:**
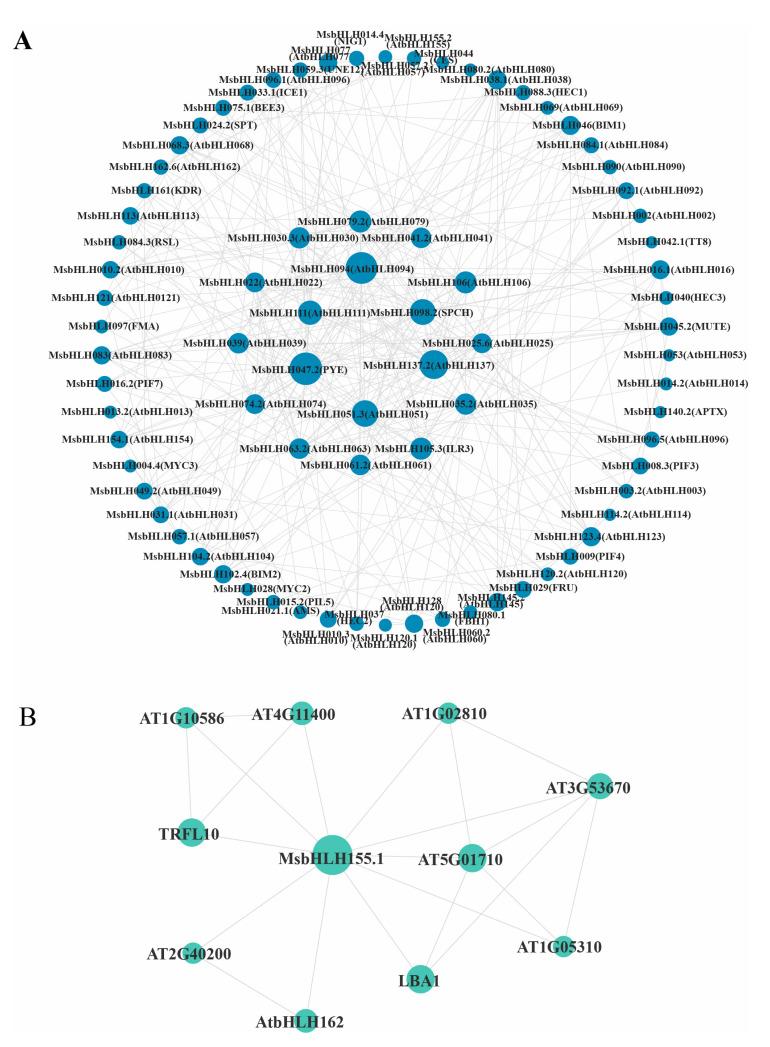
bHLH protein interaction network of *M. sieversii* based on their orthologs in *Arabidopsis*. (**A**) STRING website was used to predict the bHLH protein interaction network of *M. sieversii*. The *A. thaliana* was chosen as the reference plant, and then the minimum required interaction score (0.4000) was set to middle confidence. (**B**) Protein interaction network of MsbHLH155.1.

**Figure 7 plants-12-00620-f007:**
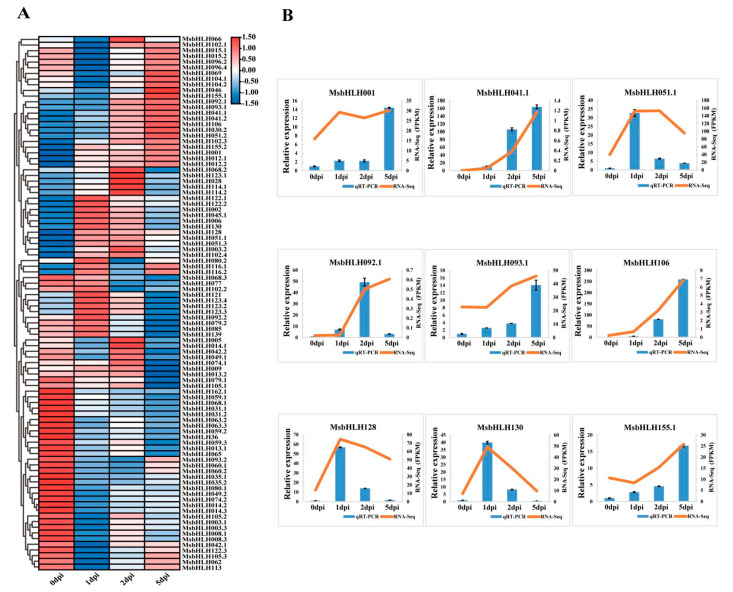
Expression analysis of DEG *MsbHLHs* upon *V. mali* infection at 0, 1, 2, 5 dpi. (**A**) Clustering expression analysis of 94 *MsbHLH* genes under *V. mali* infection based on transcriptome data by Illumina sequencing [42]. The color scale represents the fragments per kilobase million (FPKM) value after the *V. mali* infection. The values were normalized for the significant expression genes by more than 1.0 fold. Red and blue indicated high and low expression levels, respectively. (**B**) Relative expression analysis of nine *MsbHLH* genes detected by qRT-PCR. The blue column represents the FPKM values. The orange line represents the relative expression level expression profiles of the 9 *MsbHLH* genes responding to *Valsa* canker disease.

**Figure 8 plants-12-00620-f008:**
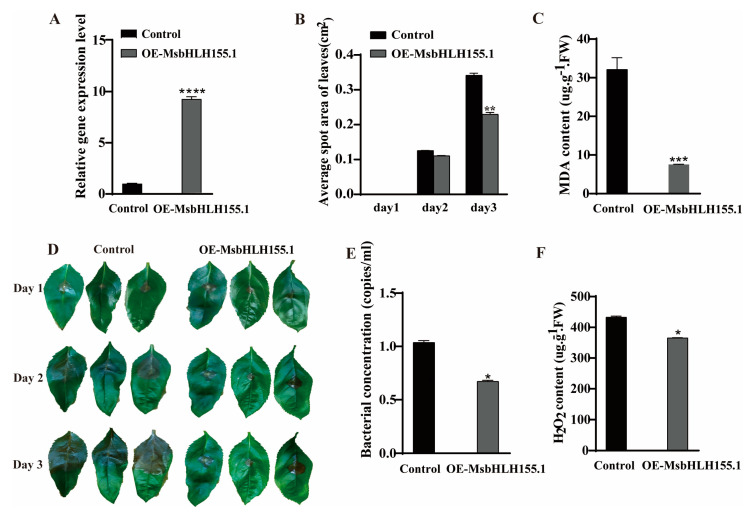
Phenotypic and physiological indexes of overexpression of *MsbHLH155.1* (*OE-MsbHLH155.1*) during canker disease. (**A**) Relative expression of *MsbHLH155.1*; (**B**,**D**) leaf lesion area after infection; (**C**) MDA content in infected leaves; (**E**) bacterial content in infected leaves; (**F**) H_2_O_2_ content in leaves infected by the pathogen. Leaves infected with p1307-flag (empty vector) as controls. (An asterisk indicates a significant difference between treatment and control plants; “*” represents *p* < 0.05; “**” represents *p* < 0.01; “***” represents *p* < 0.001; “****” represents *p* < 0.0001).

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
