# Peer review of "Genome-Wide Identification of bHLH Transcription Factor Family in Malus sieversii and Functional Exploration of MsbHLH155.1 Gene under Valsa Canker Infection"

_plants, 2023, doi:10.3390/plants12030620_

Round 1
Reviewer 1 Report
The paper submitted by Jia et al. cover an interesting topic, indeed transcription factors are multiacting proteins able to regulate various metabolic pathways in plants. The experimental design is correct and very laborious. I suggest to accept the paper but an extensive english language editing is needed in the whole text.
The abstract need to be rewritten because is not fluid and the transient transformation of MsbHLH155.1 has not been mentioned. Please reformulate it!
The keywords contain words of the title. Please change them.
Introduction
line 36: change "bHLH transcription factor" in "It"
line 49: change "There were lots of bHLH gene family has been reported in many plants" in "Several studies reported the investigation of bHLH gene family in plants,"
line 52: change "[17], Populus deltoids (185) [18], etc" in "[17] and Populus deltoids (185) [18]"
line 90-91: delete "Though many bHLH families of different species have been characterized, and im- 90 portant functions have been identified. However, no research has been reported for the 91 bHLH family in M. siseversii."
Results
line 116: change (Supplementary Table 2). in (Supplementary Table 1). and change even the supplementary tables numbering.
Please check the number of all figures and tables even the supplementary ones. There are many mistakes in the text.
The remaining text need an extensive editing of English.
Concerning the involvement of bHLH in the production of antocyanins in plants, I suggest you to cite this reference in the text:
"D’Amelia, V., Villano, C., Batelli, G., ÇobanoÄŸlu, Ö., Carucci, F., Melito, S., ... & Carputo, D. (2020). Genetic and epigenetic dynamics affecting anthocyanin biosynthesis in potato cell culture. Plant Science, 298, 110597."
Reviewer 2 Report
The manuscript titled: “Genome-wide identification of bHLH transcription factor family in Malus sieversii and function exploration of MsbHLH155.1 under the Valsa canker disease infection” concerns genetic analyses of transcription factor family bHLH in a wild apple species. The authors consider the phylogenetic relationship of the transcription factors bHLH and their role in disease resistance. The results are generally well presented and are interesting.
However, the manuscript needs correction. I have suggestions for authors listed below.
In the title the authors should write ‘MsbHLH155.1 gene’ instead of ‘MsbHLH155.1.’
In abstract, the authors should add short information what bHLH gene codes (for example in line 15) and should give the full name of bHLH.
In Introduction, the authors have devoted a great deal of text to the description of transcription factors bHLH in animals. I suggest that the authors write more about bHLH in plants while reducing the description of animal bHLH.
Line 49-50. There is a grammatical problem here.
Line 50- 52. What are numbers in parentheses? Number of bHLH genes? Please clarify in text.
Line 53-55. These two sentences have the same reference. It is enough to put the citation [19] once.
Line 61. The authors suddenly jump from information about wild fruit forests to a description of genes encoding transcription factors. What is the relationship between these topics? Somewhere in this place the authors should explain how learning about bHLH genes and bHLH proteins can help in protecting old orchards.
Lines 62-63. What are numbers in parentheses? Please clarify in text.
Line 62. Is it ‘Tianshan Wild Fruit Forest’ a proper name? If yes, it is ok. If not, please use lowercases for ‘wild fruit forest’.
Line 73. Write ‘soybean’ using a lowerecase.
Line 78. There is a place where the authors mention ‘JA’ for the first time, therefore the abbreviation must be explain here.
Line 86. ‘Jasmonic acid’ and ‘salicylic acid’ should be written using lowercases.
Line 103. Please develop the abbreviation SMART and CD-Search. , using the names of databases, software, applications etc. the authors should write it because it is not clear to the average reader what the abbreviation refers to. For example STRING database (line 214).
Lines 107-108. There is: ‘Thus 184 MsbHLH genes were named MsbHLH001- MsbHLH162.6.’ The group of 184 bHLH genes was named as ‘MsbHLH001- MsbHLH162.6’? Or the genes named using the following numbers from 001 to 162.6? It is not clear, especially as it is not a range from 1 to 184. Please clarify it.
Line 111. The sentence is not clear. There is a grammatical problem here.
Line 112. There is ‘The remaining were instability’. Instability or instable?
Line 115-116. What are BN and MN? Please develop the abbreviations. Also in supplementary table 2.
Line 252. What is DEG? Differentially expressed genes? Please explain the abbreviation in the text.
Line 275. Please explain the abbreviation MDA in the text.
Line 298. What are numbers in parentheses? Please clarify in text.
Lines 310-311. The sentence is unclear. Please rephrase it.
Line 315. There is ‘These 25 subfamilies…’ Which subfamilies? It is not clear.
Line 318- 322. The sentence is too long and unclear. Please divide it and rephrase.
Line 321. What is JAZ? Please explain the abbreviation in the text.
Line 355. What are MdERF100, MdbHLH9? Transcription factors? Please add this information in the text.
Line 364. As ‘these researches’ the authors mean the presented in the manuscript or the referenced above? It is not clear. Please clarify.
Line 372. There is ‘Sa’. It should be ‘SA’.
Line 385. What is TOPLESS? A protein? Add a word to explain.
Line 406. MsbHLH155.1 is written in italics which suggests that it is a gene while in fact it is a protein and should be without italics.
Line 416-417. Unclear. A grammatical problem.
Line 418-422. Unclear. A grammatical problem. I suggest to divide the sentence into two.
Line 486. Unclear. What does it mean that the leaves were charged?
Line 486-487. The sentence id unclear. Please rephrase it. What is ImageJ? Add a word of explanation. ImageJ software?
Figure 1 is too small and it is illegible.
Figure 2 is too small and it is not legible enough, especially the names of TFs. Phylogenetic tree of what? Please complete the information.
Figure 3. Please write the explanation what S1, S2, S3 etc. are? What are the values on X axis? bp? The names of TFs are too small and are illegible.
Figure 4. The lower scale is shifted in relation to the chromosomes.
Figure 6. line 238. There is: ‘The bHLH protein sequences in A. thaliana were chosen as the reference organism’. There is a grammatical problem. Sequences are not an organism.
Figure 7. Please explain in the caption of the table what is on the scale presented on the right of diagram in part A. Probably, it shows how many times gene expression is higher than the control but it is not written. Moreover, please explain the abbreviation FPKM.
Figure 8. Please give the values on the Y axis, that is copies/ml. Line 282. What is OE-MsbHLH155.1? add the word ‘gene’. Prefix ‘OE’ is used only in this place in the manuscript. It is confusing, because in the rest of the text ‘MsbHLH155.1’ is used. Please unify it. Besides, there is a big confusion in the caption. Please verify the captions for the parts B, C, E.
Line 285. Leave can be infected by a pathogen, not the disease. Please also add a word explaining what p1307-Flag is.
In the text, the authors refer the figures in various ways. For example Figure 4 (in line 175), Fig2 (in line 301), Fig 3 (in line 310), Fig 7B (in line 254), Fig8.C (in line 275). Please go through the whole text and unify the references to the tables.
Table S2. What are NLS, MN, BN? What are: nucl, pero, mito, plas and the numbers? Please give the explanation of it in/under the table.
Table S3. What are the numbers in columns? Please give the explanation it in/under the table.
